# Iron Inhibits the Translation and Activity of the Renal Epithelial Sodium Channel

**DOI:** 10.3390/biology11010123

**Published:** 2022-01-12

**Authors:** Abdel A. Alli, Ling Yu, Ewa Wlazlo, Sadat Kasem, Mohammed F. Gholam, Dhruv Desai, Carlos I. Lugo, Sophie Vaulont, Yogesh M. Scindia

**Affiliations:** 1Department of Physiology and Functional Genomics, College of Medicine, University of Florida, Gainesville, FL 32610, USA; aalli@ufl.edu (A.A.A.); lingyu@ufl.edu (L.Y.); mohammed.gholam@ufl.edu (M.F.G.); lugocarlos98@ufl.edu (C.I.L.); 2Division of Nephrology, Hypertension, and Renal Transplantation, Department of Medicine, College of Medicine, University of Florida, Gainesville, FL 32610, USA; sadatkasem@ufl.edu (S.K.); dhruv.desai@medicine.ufl.edu (D.D.); 3Division of Nephrology, Department of Medicine, University of Virginia, Charlottesville, VA 22904, USA; wlazlo.ewau@gmail.com; 4Department of Basic Medical Sciences, King Saud Bin Abdulaziz University for Health Sciences, Jeddah 21423, Saudi Arabia; 5Institut Cochin, Universite’ de Paris, INSERM U1016, 75014 Paris, France; sophie.vaulont@inserm.fr; 6Division of Pulmonary Critical Care and Sleep Medicine, Department of Medicine, College of Medicine, University of Florida, Gainesville, FL 32610, USA; 7Department of Pathology, College of Medicine, University of Florida, Gainesville, FL 32610, USA

**Keywords:** iron, ENaC, hypertension, salt loading, anemia

## Abstract

**Simple Summary:**

Here, we investigated the regulation of the renal epithelial sodium channel (ENaC) by iron in two established cell lines and a transgenic animal model of iron overload. We show that iron availability inhibits ENaC protein expression and its activity. We provide a discussion for putative mechanistic pathways that may be essential in this feedback inhibition.

**Abstract:**

Hypertension is associated with an increased renal expression and activity of the epithelial sodium channel (ENaC) and iron deficiency. Distal tubules absorb iron, causing perturbations that may influence local responses. In this observational study, we investigated the relationship between iron content and ENaC expression and activity using two cell lines and hepcidin knockout mice (a murine model of iron overload). We found that iron did not transcriptionally regulate ENaC in hepcidin knockout mice or in vitro in collecting duct cells. However, the renal tubules of hepcidin knockout mice have a lower expression of ENaC protein. ENaC activity in cultured Xenopus 2F3 cells and mpkCCD cells was inhibited by iron, which could be reversed by iron chelation. Thus, our novel findings implicate iron as a regulator of ENaC protein and its activity.

## 1. Introduction

The final control of sodium reabsorption occurs in the cortical collecting duct of the kidney by the epithelial sodium channel (ENaC). An increase in Na(+) transport by ENaC is thought to be a common and requisite component of salt-dependent forms of hypertension [1,2]. Several studies have shown that the regulation of ENaC by hormones including aldosterone [3], insulin [3,4,5,6], atrial natriuretic peptide (ANP) [7], endothelin-1 (ET1) [8,9,10], and arginine vasopressin (AVP) [11] are important in the fine-tuning of electrolyte balance as well as blood pressure control. The density of ENaC at the luminal membrane and its open probability is regulated by the actin cytoskeleton and scaffolding protein myristoylated alanine-rich C kinase substrate (MARCKS) [12,13,14].

Iron deficiency is common in idiopathic pulmonary arterial hypertension (IPAH) patients and is associated with increased disease severity and poor clinical outcome [15]. IPAH is associated with low iron, ferritin, and transferrin saturation levels without overt anemia, and with high hepcidin independent of the inflammatory marker interleukin-6 [16]. Dysregulation of iron metabolism is thought to be an important and independent risk factor for the onset of hypertension [16].

The hepcidin–ferroportin axis regulates systemic iron homeostasis [17]. Hepcidin, is produced in the liver and regulates steady-state iron levels by binding to cell surface ferroportin. This leads to its internalization and degradation. In mammals, Ferroportin is the only known iron export protein that releases cellular iron into circulation [17,18]. Hepcidin production is increased by mutations in hepcidin suppressors such as matriptase-2 [19,20] or more commonly settings of inflammation resulting in anemia of inflammation (previously known as anemia of chronic disease) [18,19,20]. Renal iron metabolism is a complex process involving iron import, storage, and export [21]. Little is known about iron handling by glomerular cells and most of our understanding of renal iron handling is based on studies on the tubular compartment [22,23]. Proximal tubular epithelial cells (PTECs) express ZIP8, ZIP14, and DMT1 for iron import, light and heavy chain ferritin for iron storage, and ferroportin for iron export [24]. Hence, PTECs participate in renal iron recycling [24]. We and others have shown the importance of iron regulation in PTECs in the context of acute kidney injury (AKI) and fibrosis [23,25,26]. The distal renal tubules also express the proteins ZIP8, ZIP14, and DMT1 (iron import). However, they lack the expression of light and heavy chain ferritin (iron storage), and ferroportin (iron export) [24]. As a result, distal tubules cannot recycle iron. The cortical collecting duct cells express the iron importer, transferrin receptor1 (TfR1) [27]. However, the role of iron in the pathophysiology of cortical collecting duct cells is not well established.

Considering that the cortical collecting duct cells express both TfR1 (iron import) and ENaC (sodium transport) proteins, in this study we investigated for the first time whether iron loading negatively regulates ENaC activity in cultured Xenopus distal tubule cells and mouse collecting duct cells. We also investigated changes in the protein expression of the ENaC alpha subunit and its adaptor protein MARCKS in kidneys of hepcidin deficient (*Hamp^-/-^*) mice, a model of spontaneously iron overload. We observed that iron is involved in the translational inhibition of ENaC expression and activity. Importantly, iron chelation reverted iron-induced inhibition of ENaC activity. As hypertension is associated with an increase in renal ENaC and systemic iron deficiency, our observations could have clinical implications.

## 2. Materials and Methods

### 2.1. Animals

All animal experiments were performed in accordance with the National Institutes of Health, Institutional Animal Care and Use Guidelines, and were approved by The Animal Care and Use Committee of the University of Florida. Hepcidin knockout (*Hamp^-/-^*) and their wild-type littermates (WT) (all on C57BLK background), gifts from Dr. S. Vaulont (Universite’ de Paris, INSERM), were housed and maintained in animal facilities at the University of Florida on a regular diet. The mice used in this study were approved by the University of Florida’s IACUC (protocol numbers 201810373 and 202011157).

### 2.2. Cell Culture

mpkCCD cells from male mice were maintained in DMEM and Ham’s F-12 medium (1:1 mixture) (Gibco; Grand Island, NY, USA) pH 7.4 and supplemented with 20 mM HEPES, 20 mM D-glucose, 5 µg/mL insulin, 2 mM l-glutamine, 1 nM triiodothyronine, 50 nM dexamethasone, 0.1% penicillin-streptomycin, and 2% heat-inactivated FBS. These cells were maintained at 37 °C in 5% CO_2_. To measure the effect of iron, cells were washed and treated with the above medium in 0.5% FBS in the presence or absence of 100 μM ferric ammonium citrate (FAC) for 16–18 h. Xenopus 2F3 cells were cultured in Dulbecco’s modified Eagle’s medium (DMEM)/Ham’s F-12 medium (Gibco; Grand Island, NE, USA) supplemented with 1.0% streptomycin and 0.6% penicillin (Gibco; Grand Island, NE, USA), 1.5 μM aldosterone (Sigma-Aldrich; St. Louis, MO, USA), and 5% fetal bovine serum (FBS; Gibco; Grand Island, NE, USA). These cells were maintained at 26 °C in 4% CO_2_.

### 2.3. Amiloride-Sensitive Transepithelial Current Measurements

An epithelial voltohmmeter2 equipped with chopstick electrodes (World Precision Instruments, Sarasota, FL, USA) was used to measure resistances and voltages across a confluent monolayer of Xenopus 2F3 cells treated with vehicle or FAC (100 μM). At the end of the experiment, amiloride was administered at 0.5 μM as a control.

### 2.4. Electrophysiology

A two-stage vertical puller (Narishige, Tokyo, Japan) was used to pull micropipettes from filamented borosilicate glass capillaries (TW-150F, World Precision Instruments, Sarasota, FL, USA). The pipette tip was placed on the cell surface and then negative pressure was applied using a glass syringe in order to obtain a 10- to 20-GΩ seal resistance. The pipette and bath patch recording solutions contained 150 mM NaCl, 1 mM CaCl2, 2 mM MgCl2, and 10 mM HEPES, pH 7.4. pCLAMP software (Molecular Devices, Sunnyvale, CA, USA) was used to calculate ENaC activity (i.e., number of channels in a patch [channel density (*N*)] times the open probability (*P*o) of the channel (*NP*o)). Adherent cells were treated with vehicle, FAC (100 μM), deferoxamine (DFO), a clinically used iron chelator (100 μM), or FAC + DFO (100 μM each). This dose of FAC [22] and DFO [28] has been used on renal epithelial cells and has not been found to be toxic.

### 2.5. Immunohistochemistry and Immunofluorescence

Formalin-fixed paraffin-embedded kidney tissue sections were deparaffinized by sequential changes of xylene (2 changes) (Fisher Scientific; Pittsburgh, PA, USA), followed by 2 exchanges of 100% ethanol, 1 exchange of 95% ethanol, 1 exchange of 70% ethanol, 1 exchange of 50% ethanol, and then 1 exchange of type 1 water for 3 min intervals each. The slides were then boiled in citrate buffer for 20 min (Vector Laboratories, Inc.; Burlingame, CA, USA) and washed for 3-min in type 1 water and then in 1X phosphate-buffered saline (1XPBS) (Corning; Manassas, VA, USA) for 5 min. After blocking with 2.5% normal horse serum (Vector Laboratories, Inc.) for 20 min the tissues were incubated for 60 min with a 1:500 dilution of primary rabbit polyclonal anti-MARCKS antibody (ab72459; Abcam, Boston, MA, USA) followed by a 60-min incubation with mouse anti-ENaC alpha antibody (StressMarq Biosciences Inc., Victoria BC, CA, Canada; SMC-242D), mouse anti-ENaC beta antibody (StressMarq Biosciences Inc., Victoria BC, CA, Canada; SMC-240D) or anti-ENaC gamma 2102 antibody conjugated with the Dylight NHS ester 594 kit (ThermoFisher, Waltham, MA, USA). After 3 washes with PBS, the tissues were incubated with VectaFluor Duet Reagent (Vector Laboratories, Inc.) for 30 min at room temperature. After 3 washes with PBS, the slides were mounted with Vectashield anti-fade mounting media (Vector Laboratories, Inc.). The slides were imaged for fluorescence using a Nikon TE microscope at 40X.

### 2.6. Real Time PCR

Frozen tissue or fresh cells were re-suspended in RLT buffer (Qiagen Inc., Valencia, CA, USA) and homogenized using the TissueLyser system (Qiagen). RNeasy Plus mini kit (Qiagen) was used to purify the RNA. Using 1 μg of RNA, cDNA was synthesized (iScript cDNA synthesis kit, Bio-Rad Laboratories, Hercules, CA, USA). The cDNA template was mixed with iTAQ SYBR green universal supermix (Bio-Rad) and predesigned primers for ENaC alpha (Bio-Rad). Quantitative PCR was performed using a CFX Connect System (Bio-Rad). PPIA (Bio-Rad) was used as the reference gene. Data are expressed as fold change over control and were calculated using the 2^−ΔC(T)^ method.

### 2.7. Perls Detectable Iron Staining

Twelve-week-old WT or *Hamp^-/-^* (on C57BLK background) were euthanized. Kidneys were perfusion fixed in formalin, paraffin-embedded, and cut into 5-micron sections. After bringing them to PBS by sequential washes of xylene and alcohol gradients, sections were stained with Perl’s detectable iron kit (Sigma, St. Louis, MO, USA) as per manufacturer’s instructions. After washing excess reagent, tissue was counter-stained with nuclear fast red and imaged for blue iron deposits using a Nikon TE microscope.

### 2.8. Urinary and Serum Iron Measurements

Twelve-week-old WT or *Hamp^-/-^* (on C57BLK background) were housed individually for 24 h in metabolic cages for urine collection. Mice had free access to food and drinking water. At the end of the study, urine was frozen at −80 °C until further use. Mice were anesthetized using isofluorane. Blood was collected via cardiac puncture and the mice were euthanized. After waiting for 30 min, the blood was spun at 5000 rpm for 20 min to separate serum. Urinary and serum iron levels were measured using the iron assay kit (Sigma, St. Louis, MO, USA) as per the manufacturer’s instructions.

### 2.9. Measurement of Urinary Electrolytes

The concentrations of urinary sodium and potassium from spot urinary collections were measured using a SmartLyte Electrolyte ISE Analyzer (Diamond Diagnostics, Holliston, MA, USA). Briefly, urine samples were subject to centrifugation at 13,000 rpm for 6 min before being diluted 1:10 in urine diluent (Diamond Diagnostics) and mixed by vortexing for 3 s. The samples were then run in urine mode.

### 2.10. Blood Pressure Measurements

Blood pressure measurements were taken by the tail-cuff method (IITC MRBP System from Life Science Inc., Woodland Hills, CA, USA). Data were analyzed using Version 1.63 of the MRBP Software.

### 2.11. Statistical Analysis

Statistical significance was determined using a 2-tailed unpaired Students *t*-test. A Mann-Whitney test was used for groups not passing the normality test. Two-way analysis of variance (ANOVA) was used to compare more than 2 groups of experimental conditions. All analyses were performed using GraphPad Prism 9 (GraphPad Inc., San Diego, CA, USA) For the transepithelial current measurements, statistical analysis was performed using SigmaPlot software (Systat, San Jose, CA, USA). A Student’s *t*-test was used to compare the two groups. Data were reported as means ± SEM and a *p*-value < 0.05 was considered statistically significant.

## 3. Results

### 3.1. Hamp^-/-^ Mice Are Iron Overloaded and Accumulate Iron in the Renal Tubules but Not the Glomeruli Compared to WT Controls

*Hamp^-/-^* or WT controls were kept on a regular rodent diet and euthanized at 12 weeks of age. Compared to WT controls, *Hamp^-/-^* mice had significantly elevated iron levels in the serum and urine (Figure 1A,B). Since high urinary iron secretion is an indicator of high renal iron content, we stained the kidneys of these mice for iron. We could not observe Perl’s detectable iron in the WT control mice. However, the kidneys of *Hamp^-/-^* mice were severely iron overloaded as seen by the blue iron deposits. Remarkably, the iron deposits were seen mostly in the tubular segment of the nephron, whereas the glomeruli were devoid of such deposits (Figure 1C,D).

### 3.2. Iron Does Not Transcriptionally Regulate ENaC Alpha Expression in Mice or A Cortical Collecting Duct Cell Line

To elucidate whether the observed lower protein expression and activity of ENaC alpha in iron overloaded mice and cells is due to its transcriptional control by iron, we measured ENaC alpha gene expression in WT mice, *Hamp^-/-^* mice, and in iron-loaded mpkCCD cells (a cortical collecting duct cell line). We found a non-significant but lower expression of ENaC alpha gene expression in the *Hamp^-/-^* mice (Figure 2A). To validate if iron directly regulates the gene transcription of ENaC alpha in ENaC alpha expressing mpkCCD cells (a cortical collecting duct cell line), we treated these cells with serum-free medium or medium containing 100 μM ferric ammonium citrate (FAC) for 12 h. We found that iron treatment did not transcriptionally regulate ENaC alpha in mpkCCD cells (Figure 2B). There was increased oxidative stress in iron-treated cells as indicated by a significant increase in NAD(P)H Quinone Dehydrogenase 1 (*Nqo1*) (Figure 2C). However, the cell death program was not activated by FAC at the given concentration (Figure 2D) and agrees well with published literature [22].

### 3.3. ENaC Protein Expression Is Reduced in the Kidneys of Iron Overloaded Hamp^-/-^ Mice

To determine whether iron loading in the native kidney influences the protein expression of ENaC and its adaptor protein MARCKS, we performed immunohistochemistry on paraffin-embedded kidneys from wild-type mice and iron-loaded *Hamp^-/-^* mice. ENaC alpha subunit protein expression was dramatically reduced in the *Hamp^-/-^* mice compared to the wild-type mice (Figure 3). MARCKS protein expression was also greater in the wild-type mice compared to the *Hamp^-/-^* (Figure 3).

We also investigated the protein expression of renal ENaC beta subunit and MARCKS expression since ENaC functions most efficiently as a complex of alpha, beta, and gamma subunits. Like, ENaC alpha, protein expression of ENaC beta is shown to be greater in the WT mice compared to the *Hamp^-/-^* mice (Figure 4). Additionally, MARCKS protein expression is shown to be greater in the WT mice compared to the *Hamp^-/-^* mice (Figure 4).

Like ENaC alpha and ENaC beta protein expression, the protein expression of the ENaC gamma subunit is shown to be greater in the WT mice compared to the *Hamp^-/-^* mice (Figure 5). Additionally, less MARCKS protein expression can be seen in the *Hamp^-/-^* mice compared to the wild-type mice (Figure 5).

### 3.4. ENaC Activity in Cultured Xenopus 2F3 Cells Is Inhibited by Iron

First, we investigated whether ENaC activity in Xenopus 2F3 cells is altered by iron loading in these cells. The cells were treated with FAC or vehicle for different time points and transepithelial voltage and resistance were measured before calculating transepithelial current using Ohms law. As shown in Figure 6, FAC treatment decreased amiloride-sensitive transepithelial current after 90 min of treatment.

### 3.5. ENaC Activity in Cultured Mouse Cortical Collecting Ducts Cells Is Inhibited by Iron Loading

Next, we investigated for the first time whether ENaC activity in the mouse collecting duct is regulated by iron loading using an established and commonly used cellular model for the investigation of ENaC regulation. mpkCCD cells were treated with FAC for 240 min before performing single-channel patch-clamp studies using the cell-attached configuration. ENaC activity (NPo) was lower (*p* = 0.007) in the FAC treated group compared to the vehicle group (Figure 7A). This effect of FAC (iron) was negated by adding Deferoxamine (DFO), a clinically used iron chelator [29].

### 3.6. The Number of Patches with ENaC Activity but Not Its Open Probability Is Attenuated after Iron Loading

Next, we measured the number of channels in each patch along with the open probability of the channel. The number of patches with active channels decreased (*p* = 0.013) in FAC treated cells compared to cells treated with vehicle (Figure 7B). The FAC-induced inhibition of active channels was reversed by DFO (Figure 7B). There was no appreciable difference in the open probability of ENaC between the FAC treated cells and the vehicle-treated cells (Figure 7C). The conductance of the channels measured in the mpkCCD cells was 17 pS (Figure 7E) and the single-channel characteristics showed long open and closed times.

### 3.7. Base-Line Systolic Blood Pressure Is Lower in Hamp^-/-^ Mice Compared to Wild-Type Mice

ENaC plays an important role in total body sodium balance and blood pressure control. Therefore, we investigated whether the inhibition of ENaC protein expression and activity correlates with changes in urinary sodium concentration and a decrease in systolic blood pressure in *Hamp^-/-^* iron overloaded mice. At baseline level, there was no significant difference in the urinary spot sodium (WT 107.8 ± 2.73 vs. KO 92.05 ± 5.02) and potassium (WT 67.93 ± 4.76 vs. KO 60.45 ± 5.76) levels. However, as shown in Figure 8, the baseline systolic blood pressure of *Hamp^-/-^* mice was significantly lower compared to wild-type litter mate controls.

## 4. Discussion

Recent evidence suggests that the proximal tubule is not the only site of iron uptake in the nephron. Moulouel et al. showed that filtered plasma iron is taken up at the distal nephron [30]. Here, we investigated for the first time the regulation of ENaC and MARCKS in the distal tubule and cortical collecting duct of cultured cells and the native kidney of WT and iron-loaded mice.

*Hamp^-/-^* mice are a well-established animal model that, under normal dietary conditions, accumulate iron in multiple organs including the kidneys [26,31] As shown above, these mice accumulate iron mostly in the renal tubules. The technically harsh conditions involved in staining Perl’s detectable iron did not permit co-staining and identification of individual segments of the tubules that were iron-loaded. However, all major renal tubular epithelial cells including the proximal, distal, and collecting duct cells take up iron [26,29]. While we did not observe transcriptional inhibition of ENaC in the iron overloaded kidneys, its translation was suppressed by iron. Iron can bind to iron regulatory proteins (IRPs), leading to the dissociation of IRPs from the iron-responsive elements IRE and alter the translation of target transcripts [29]. It is not yet known whether iron exerts such translational control of ENaC.

The expression and activity of ENaC in the kidney are positively regulated by anionic phospholipids phosphates (e.g., phosphatidylinositol bisphosphates) and the adaptor/scaffolding protein MARCKS [12,13,14]. Although ENaC directly binds to phosphatidylinositol bisphosphates (PIP2) at the luminal plasma membrane, both ENaC and PIP2 are rare. MARCKS serves as an adaptor protein to sequester PIP2 and increase the local concentration close to ENaC to allow for its regulation. The regulation of ENaC by PIP2 involves the subcellular localization of MARCKS. MARCKS regulates ENaC at the level of N, the density at the luminal membrane, and Po, the open probability of the channel.

Here, we investigated the regulation of FAC treatment on ENaC activity in cultured Xenopus 2F3 distal tubule cells and mouse cortical collecting duct cells. The effect of acute FAC treatment was the inhibition of ENaC activity in both cell types. Similarly, we observed a decrease in ENaC protein expression in the native kidneys of iron overloaded *Hamp^-/-^* mice compared to wild-type mice. The co-localization of ENaC with MARCKS was also shown to be greater for the WT mice compared to the *Hamp^-/-^* mice. This suggests that iron is presumably causing the translocation of MARCKS from the membrane to the cytoplasm. We have previously demonstrated that MARCKS translocation from the membrane to the cytoplasm in ENaC expressing distal tubule and collecting duct cells occurs in a protein kinase C (PKC), Ca^2+^/calmodulin, and protease dependent manner. For this study, our focus was to investigate the iron-dependent regulation of ENaC and MARCKS in the kidney. A previous study by Turi et al. showed intracellular iron accumulation in airway epithelial cells requires concurrent sodium/potassium exchange [32]. Even though MARCKS is expressed in both tissues, the regulation of ENaC between the kidney and lung is different. First, there are differences in the proteases and isoforms of PKC expressed in the kidneys and lungs. Second, the stoichiometry of ENaC subunits, conductance, and single-channel characteristics of the channel may be different in the kidney and lung.

To determine whether the reduced expression and activity of renal ENaC in *Hamp^-/-^* mice contributes to a blood pressure phenotype, we measured their systolic blood pressure. Although we did not observe significant changes in urinary sodium and potassium concentrations between the two groups, the *Hamp^-/-^* mice have significantly lower baseline systolic blood pressure compared to their WT littermate controls. This suggests the negative feedback regulation that iron has on renal ENaC is accompanied by attenuation in blood pressure.

Our study has a few limitations. We did not investigate the effects of chronic treatment of iron in the distal tubules or the collecting ducts. A second limitation is that we did not investigate the regulation of other actin cytoskeleton proteins which are known to regulate MARCKS and ENaC. Finally, we did not investigate regional differences in ENaC alpha, beta, and gamma subunits protein expression between the wild-type and *Hamp^-/-^* groups. The main reason for choosing the renal cortex to investigate ENaC protein expression between the two groups is that we included a mouse cortical collecting duct cell line in this study to measure ENaC activity by electrophysiology. Differential protein expression of ENaC subunits between the cortex, inner medulla, and outer medulla warrant further investigation. While we provide evidence that iron can regulate ENaC, studies to tease out the mechanism by which iron regulates ENaC under physiological and pathophysiology conditions are warranted.

## 5. Conclusions

Our study identifies an unappreciated role of iron in the regulation of ENaC which can affect sodium homeostasis and blood pressure. The observations that hepcidin deficiency and renal iron overload are associated with a lower systolic blood pressure supports our ongoing hypothesis that hepcidin-induced chronic anemia of inflammation may contribute to hypertension by dysregulating ENaC activity and renal sodium retention.

**Clinical Perspective:** As discussed above, hypertension is associated with increased ENaC expression. Anemia is common and is associated with impaired clinical outcomes in chronic kidney disease (CKD). This anemia is driven by hepcidin, a hepatic hormone. CKD of multiple etiologies is associated with high hepcidin concentrations driven by inflammation and in part by decreased clearance of hepcidin by the diseased kidneys [33,34,35]. While studies have established the role of hepcidin in pulmonary hypertension, hepcidin-induced anemia (iron deficiency) could contribute to renal ENaC activity and hence play a critical role in regulating hypertension. Our descriptive findings are consistent with a hypothetical model depicted in Figure 9 and could pave way for novel research to investigate this testable hypothesis and identify a druggable target to mitigate a global public health problem.

During the progression of CKD, inflammation and impaired renal hepcidin clearance can lead to ferroportin degradation and iron sequestration within the spleen and liver. Reduced iron availability increases/stabilizes ENaC protein and its activity, mechanisms of which are still unknown. Increased expression and activity of ENaC potentiates hypertensive CKD.

## Figures and Tables

**Figure 1 biology-11-00123-f001:**
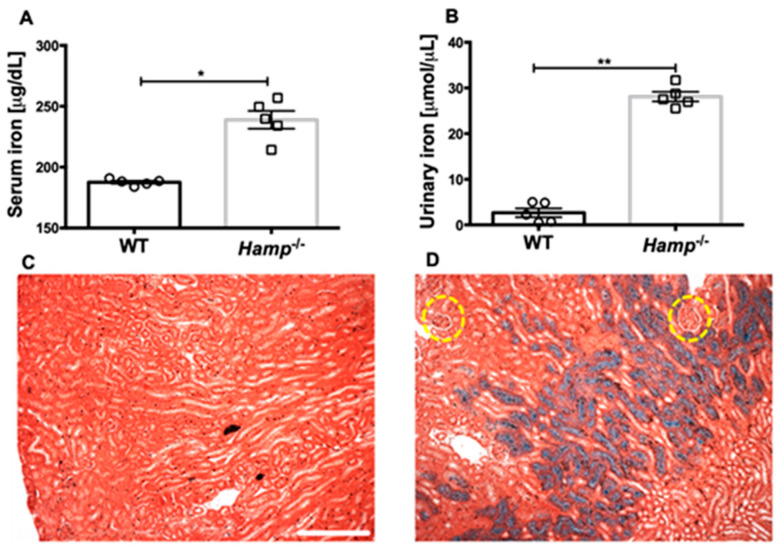
Iron overloaded phenotype of *Hamp^-/-^* mice. *Hamp^-/-^* mice kept on a regular rodent diet contain high iron in the serum (**A**) and urine (**B**). Excessive iron deposits are observed in the renal tubules of *Hamp^-/-^* mice. Kidneys of 12-week-old WT controls (**C**) or *Hamp^-/-^* mice (**D**) were stained for Perl’s detectable iron. WT kidneys did not show any Perl’s detectable iron deposits. However, excessive iron deposits (blue) were detected mostly in the renal tubules. The glomeruli (circled yellow) were devoid of these iron deposits. Perl’s detectable iron is blue. The counterstain is red. Scale bar is 100 μm. Data from 5 mice are shown and statistical significance was determined by a 2-tailed Mann–Whitney test and are plotted as mean ± SEM. * *p* < 0.05, ** *p* < 0.001.

**Figure 2 biology-11-00123-f002:**
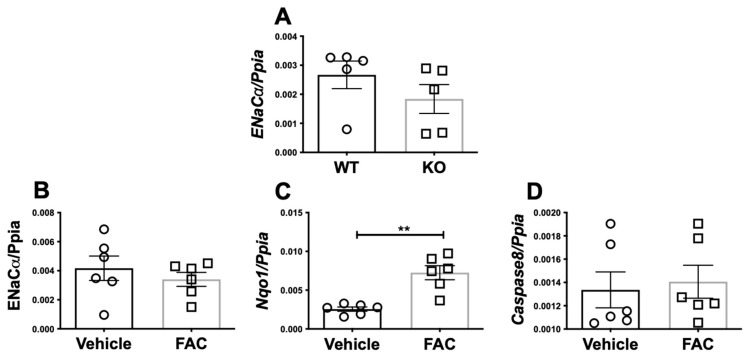
Iron does not transcriptionally regulate ENaC alpha. The kidneys of 12-week-old WT or iron overloaded *Hamp^-/-^* mice were analyzed by real time PCR for gene expression of ENaC alpha. There was no significant difference in ENaC alpha gene expression of WT or spontaneously iron overloaded *Hamp^-/-^* mice under basal conditions (**A**). We confirmed that iron can regulate ENaC alpha gene transcription in mpkCCD cells, a cortical collecting duct cell line that is known to express ENaC alpha. 200 × 10^5^ cells were serum-starved for 12 h and then cultured for 12 more hours in vehicle or FAC (100 uM). RNA was purified and then cDNA was synthesized. ENaC alpha gene expression was measured. Iron loading did not regulate ENaC alpha gene expression in mpkCCD cells (**B**), though it increased oxidative stress (**C**). One hundred uM FAC did not activate cell death program under the experimental conditions (**D**). Data from 2 independent experiments were pooled and analyzed. The statistical significance was determined by a 2-tailed Mann–Whitney test and is plotted as mean ± SEM. ** *p* < 0.001.

**Figure 3 biology-11-00123-f003:**
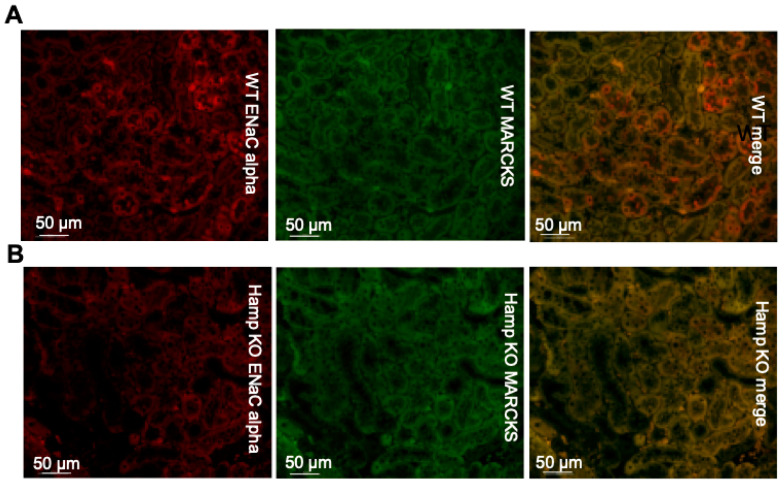
Immunohistochemistry analysis of ENaC alpha and MARCKS protein expression in *Hamp^-/-^* (KO) mice or wild-type (WT) mice. (**A**) ENaC alpha (red) protein expression and MARCKS (green) protein expression in WT control mice. (**B**) ENaC alpha (red) protein expression and MARCKS (green) protein expression in iron overloaded *Hamp^-/-^* mice. Data is representative of 3 mice in each group.

**Figure 4 biology-11-00123-f004:**
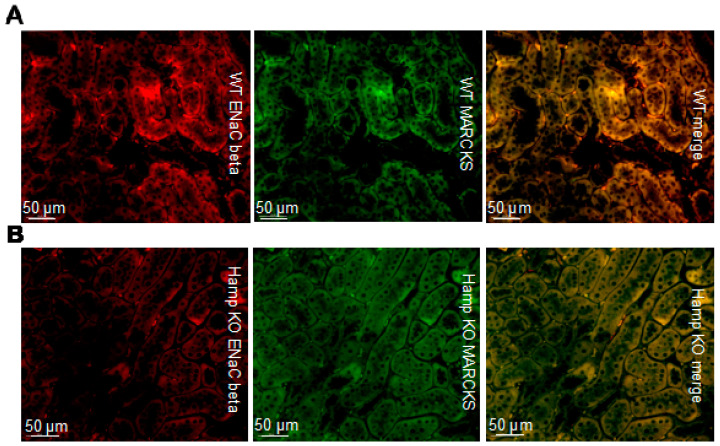
Immunohistochemistry analysis of ENaC beta and MARCKS protein expression in *Hamp^-/-^* (KO) mice or wild-type (WT) mice. (**A**) ENaC beta (red) protein expression and MARCKS (green) protein expression in WT controls. (**B**) ENaC beta (red) protein expression and MARCKS (green) protein expression in iron overloaded *Hamp^-/-^* mice. Data is representative of 3 mice in each group.

**Figure 5 biology-11-00123-f005:**
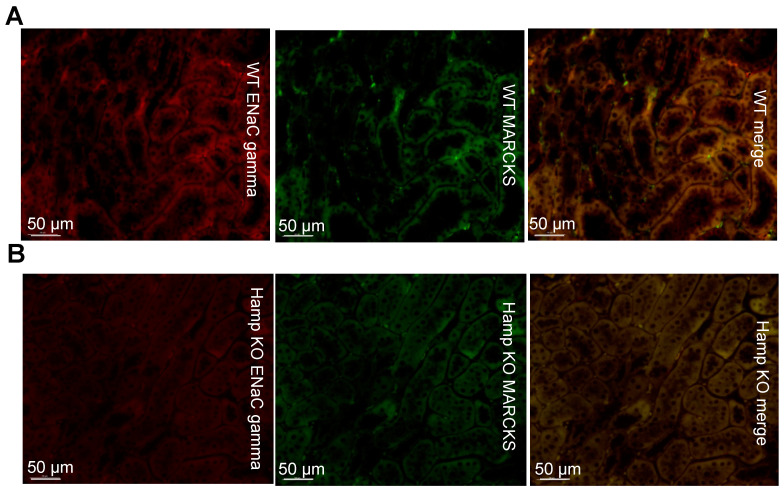
Immunohistochemistry analysis of ENaC gamma and MARCKS protein expression in *Hamp^-/-^* (KO) mice or wild-type (WT) mice. (**A**) ENaC gamma (red) protein expression and MARCKS (green) protein expression in WT controls. (**B**) ENaC gamma (red) protein expression and MARCKS (green) protein expression in iron overloaded *Hamp^-/-^* mice. Data is representative of 3 mice in each group.

**Figure 6 biology-11-00123-f006:**
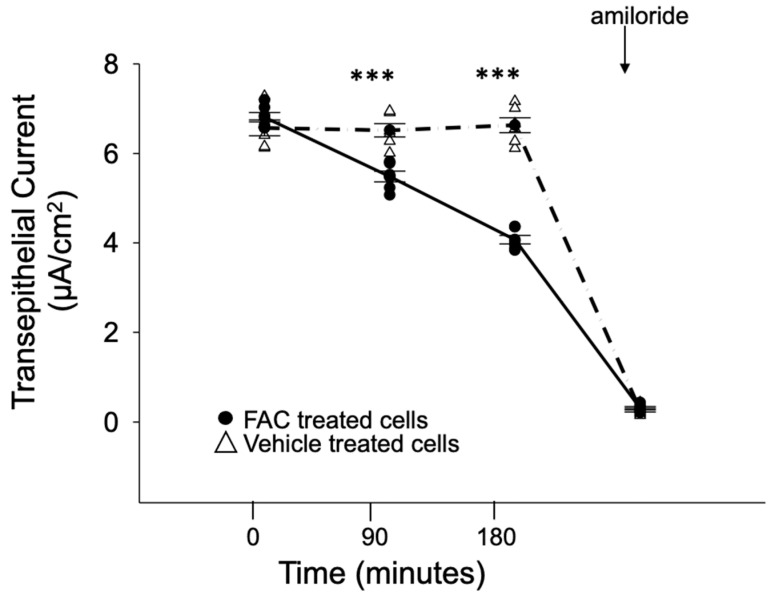
Iron decreases amiloride-sensitive transepithelial current in Xenopus 2F3 cells. Xenopus 2F3 cells were treated with either FAC or vehicle for different time points. At the end of the experiment, the ENaC inhibitor amiloride was added as a control to show the current was amiloride-sensitive. *n* = 6 permeable supports per group. A Student-*t*-test was performed to compare the groups. *** represents *p*-value < 0.001.

**Figure 7 biology-11-00123-f007:**
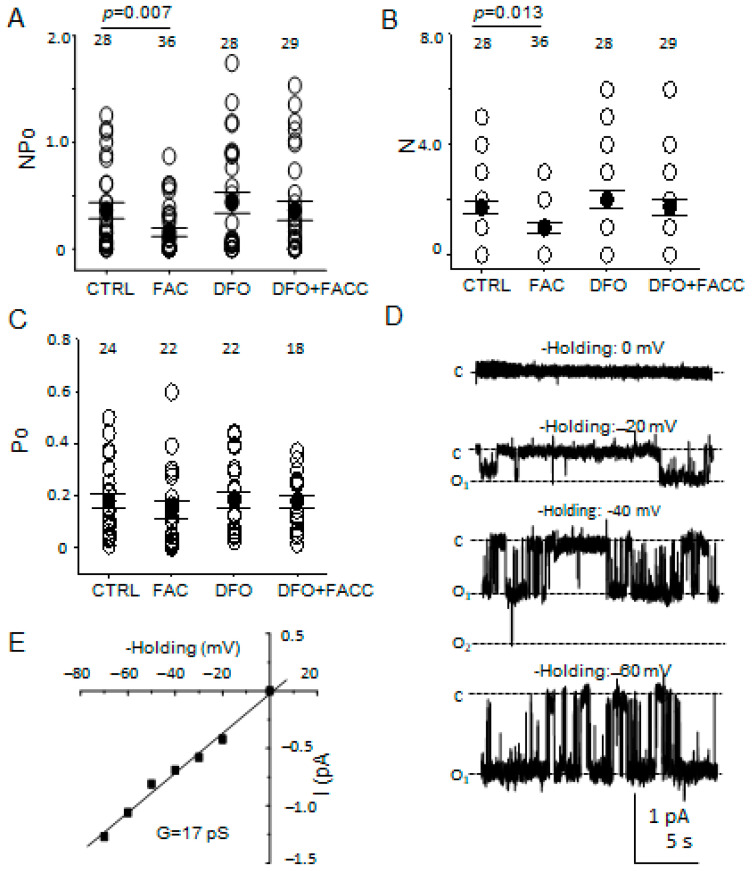
Single-channel patch-clamp studies measuring ENaC activity in mpkCCD cells treated with FAC, DFO, or DFO and FAC. (**A**) ENaC activity was calculated and shown as NPo. (**B**) The number of active channels in a patch is given as N. (**C**) The open probability of the channel is given as Po. (**D**) Representative traces show the channels have long open and closed times. C represents the channel in a closed state while O represents the open state of a channel. (**E**) I-V curve shows the conductance of the channels. The number of patches in each group is given at the top of each column.

**Figure 8 biology-11-00123-f008:**
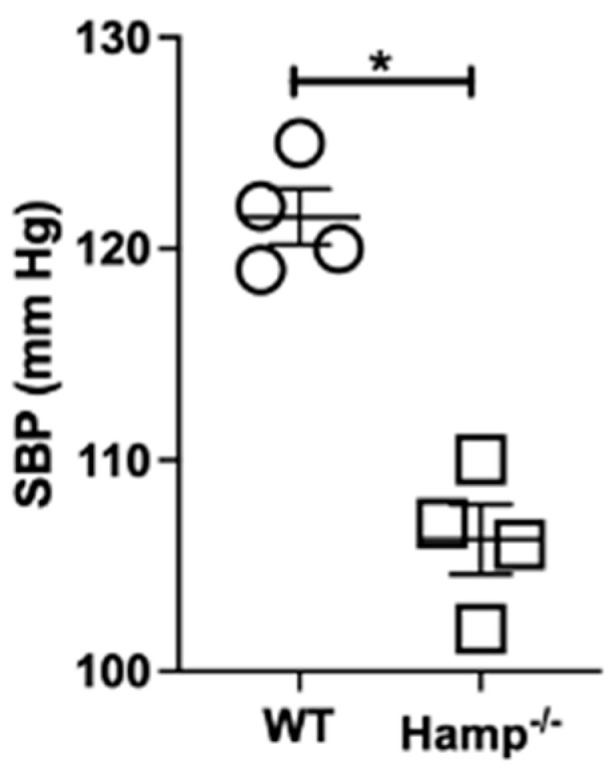
Base-line blood pressure measurements in wild-type and *Hamp^-/-^* mice. Systolic blood pressure measurements in each group were measured by the tail-cuff method. *n* = 4 mice in each group. The statistical significance was determined by a 2-tailed Mann–Whitney test and is plotted as mean ± SEM. * *p* < 0.05.

**Figure 9 biology-11-00123-f009:**
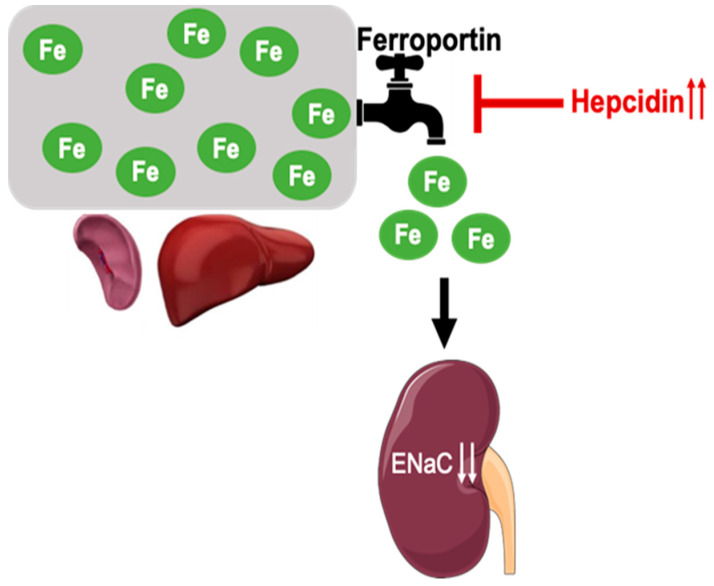
Hepcidin induced iron deficiency and hypertensive kidney disease: A hypothetical model.

## Data Availability

Not applicable.

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
