# Peer review of "Iron Inhibits the Translation and Activity of the Renal Epithelial Sodium Channel"

_biology, 2022, doi:10.3390/biology11010123_

Round 1

Reviewer 1 Report

This study discusses the role of iron in regulation of the renal sodium channel activity. The results have clinical implications w.r.t. hypertension. Overall the manuscript is very well written, concepts articulated well even to a non-expert. I only have a few comments to improving:

1. Please proof reading the manuscript as there are several places with grammatical errors, sentence corrections required, etc.

2. For Fig. 2: The authors read NqoI levels as a measure of oxidative stress after FAC treatment. How does this relate to transcriptional control? Or were you looking at post-transcriptional effects after FAC treatment as I believe oxidative stress is known to have an impact there?

Fig. 4: Perhaps you can plot this differently or have distinct legends? As is, its is difficult to read the different treatment groups as I can see closed and open bullets on both trendlines. 

Author Response

The authors would like to thank both the reviewer for taking the time to carefully review our manuscript and providing us with helpful suggestions. We have performed additional experiments and added additional data in this revised manuscript in order to address each Reviewer comment. We are grateful for the opportunity to be able to submit a revised manuscript.

Reviewer 1 Comments

This study discusses the role of iron in regulation of the renal sodium channel activity. The results have clinical implications w.r.t. hypertension. Overall the manuscript is very well written, concepts articulated well even to a non-expert. I only have a few comments to improving:

1. Please proof reading the manuscript as there are several places with grammatical errors, sentence corrections required, etc

We have proof read and corrected the grammatical and spelling mistakes.

2. For Fig. 2: The authors read NqoI levels as a measure of oxidative stress after FAC treatment. How does this relate to transcriptional control? Or were you looking at post-transcriptional effects after FAC treatment as I believe oxidative stress is known to have an impact there?

That is correct, we are investigating post-transcriptional effects after FAC treatment in which oxidative stress is known to play a role in ENaC regulation.

3. Fig. 4: Perhaps you can plot this differently or have distinct legends? As is, its is difficult to read the different treatment groups as I can see closed and open bullets on both trendlines. 

As suggested, we replotted Figure 4 (new Figure 6) to include distinct line graphs and symbols. 

Reviewer 2 Report

In the present study, the authors performed a series of experiments to prove the Fe-conducted ENaC activity and indicated the iron plays a potential role in blood pressure/sodium regulation. It's novel and may have a clinical perspective. however, several points are needed to be further described and clarified.

  1. The ENaC locates comprehensively on both cortical and medullary collecting duct tubules. if the authors focus on the cortex, the reason or the difference between such different areas should be mentioned.
  2. Since the final target of this study is ENaC, which plays a critical role in distal sodium regulation. The sodium regulation should also be investigated in vivo. The present study provided the serum and urinary iron, but there is no evidence about sodium.
  3. The doses or concentration of each compound should have references or there require the dose-dependence curves to support the dose/concentration is within regular physiological ranges.
  4. The ENaC should contain all three subunits because either the gene KO or FAC had no effects on the gene expression. Also, the in vivo protocol should have protein expression to support chronic function.
  5. The IHC for the ENaC and MARCKS is not convincing. the picked areas are not equivalent. The high expression of MARCKS in WT was picked but not in the KO mice. In KO mice, there was another area indicating the positive areas but were not chosen. The representative images should be rearranged.
  6. Figure 5 showed the ENaC changes in NP0 are mainly due to the changes in the N, the change of density. The measurement for the marker's density should be specified.
  7. If the authors want to imply the perspective in kidney/hypertension, the blood pressure data or the investigation of the regulatory factors are critical to the conclusion. 

Author Response

Reviewer 2 Comments

In the present study, the authors performed a series of experiments to prove the Fe-conducted ENaC activity and indicated the iron plays a potential role in blood pressure/sodium regulation. It's novel and may have a clinical perspective. however, several points are needed to be further described and clarified.

The authors would like to thank the reviewer for taking the time to carefully review our manuscript and providing us with helpful suggestions. We have performed additional experiments and added additional data in this revised manuscript in order to address the comment. We are grateful for the opportunity to be able to submit a revised manuscript.

  1. The ENaC locates comprehensively on both cortical and medullary collecting duct tubules. if the authors focus on the cortex, the reason or the difference between such different areas should be mentioned.

We realize that ENaC is expressed in the cortex, outer medulla, and inner medulla, but in this manuscript, we focused on its expression in the cortex.  As suggested, we added a few sentences to the last paragraph of the discussion section to explain reasons for focusing on ENaC expression in the cortex and we state that we plan to investigate regional differences of its expression within iron overloaded mice in a follow-up manuscript.

  1. Since the final target of this study is ENaC, which plays a critical role in distal sodium regulation. The sodium regulation should also be investigated in vivo. The present study provided the serum and urinary iron, but there is no evidence about sodium.

As suggested, we performed additional experiments and now include data (Figure 8) on urinary sodium from both groups of mice.

  1. The doses or concentration of each compound should have references or there require the dose-dependence curves to support the dose/concentration is within regular physiological ranges.

We have addressed this comment by the reviewer, by providing we appropriate references (31, 32) in the revised manuscript.

  1. The ENaC should contain all three subunits because either the gene KO or FAC had no effects on the gene expression. Also, the in vivo protocol should have protein expression to support chronic function.

As suggested, we performed additional experiments and now include data (new Figure 3, 4, 5) for all three ENaC subunits at the protein level for each group of mice.

  1. The IHC for the ENaC and MARCKS is not convincing. the picked areas are not equivalent. The high expression of MARCKS in WT was picked but not in the KO mice. In KO mice, there was another area indicating the positive areas but were not chosen. The representative images should be rearranged.

As suggested, we now show the entire representative IHC images for each group in order to better show areas that are equivalent.  This is now shown in the new figures 3, 4,5.

  1. Figure 5 showed the ENaC changes in NP0 are mainly due to the changes in the N, the change of density. The measurement for the marker's density should be specified.

Yes, the change in N means change in ENaC density, and this N is directly counted as the number of channels (units of current) recorded by patch clamp in the cell attached configuration. In this recording, a glass pipette is used to patch a small piece of plasma member (usually, there are few channels within a patch), and each opened channel is recorded as a unit of current. For example, in fig 5D, at a holding membrane potential of -40 mV, two units of current were recorded.  The opening or closing of each channel is random, given ENaC is slow, has long open times, and all channels are likely to be open spontaneously within 8-10 minutes of a recording period. In our experiment, we usually record around 20-25 minutes for each patch.

  1. If the authors want to imply the perspective in kidney/hypertension, the blood pressure data or the investigation of the regulatory factors are critical to the conclusion. 

As suggested, we performed additional experiments and we now show blood pressure data (Figure 8) from each of the two groups.

Round 2

Reviewer 2 Report

The authors' followed up data have met my requirement. Despite the IHC data is only semi-quantified, I think the present data can prove the conclusion.